**Knowledgebase & Database Resources**

# Echinobase: a resource to support the echinoderm research community

Cheryl A. Telmer [ID],[1] Kamran Karimi [ID],[2] Macie M. Chess,[1] Sergei Agalakov,[2] Bradley I. Arshinoff,[2] Vaneet Lotay,[2] Dong Zhuo Wang,[2] Stanley Chu,[2] Troy J. Pells,[2] Peter D. Vize,[2] Veronica F. Hinman,[1] Charles A. Ettensohn[1],*

[1]Department of Biological Sciences, Carnegie Mellon University, Pittsburgh, PA 15213, USA
[2]Department of Biological Sciences, University of Calgary, Calgary, AB, Canada T2N 1N4

*Corresponding author: Department of Biological Sciences, Carnegie Mellon University, 4400 5th Ave., Pittsburgh, PA 15213, USA. Email: ettensohn@cmu.edu

Echinobase (www.echinobase.org) is a model organism knowledgebase serving as a resource for the community that studies echinoderms, a phylum of marine invertebrates that includes sea urchins and sea stars. Echinoderms have been important experimental models for over 100 years and continue to make important contributions to environmental, evolutionary, and developmental studies, including research on developmental gene regulatory networks. As a centralized resource, Echinobase hosts genomes and collects functional genomic data, reagents, literature, and other information for the community. This third-generation site is based on the Xenbase knowledgebase design and utilizes gene-centric pages to minimize the time and effort required to access genomic information. Summary gene pages display gene symbols and names, functional data, links to the JBrowse genome browser, and orthology to other organisms and reagents, and tabs from the Summary gene page contain more detailed information concerning mRNAs, proteins, diseases, and protein–protein interactions. The gene pages also display 1:1 orthologs between the fully supported species *Strongylocentrotus purpuratus* (purple sea urchin), *Lytechinus variegatus* (green sea urchin), *Patiria miniata* (bat star), and *Acanthaster planci* (crown-of-thorns sea star). JBrowse tracks are available for visualization of functional genomic data from both fully supported species and the partially supported species *Anneissia japonica* (feather star), *Asterias rubens* (sugar star), and *L. pictus* (painted sea urchin). Echinobase serves a vital role by providing researchers with annotated genomes including orthology, functional genomic data aligned to the genomes, and curated reagents and data. The Echinoderm Anatomical Ontology provides a framework for standardizing developmental data across the phylum, and knowledgebase content is formatted to be findable, accessible, interoperable, and reusable by the research community.

Keywords: echinoderm; model organism; knowledgebase; genomics; database; ontology; biocuration

## Introduction

Echinoderms, a phylum within the deuterostomes that includes sea urchins, sea cucumbers, and sea stars, have been used as model organisms for biological research for over 100 years. Adult echinoderms are abundant, easy to obtain, and inexpensive to maintain in the laboratory. These animals can be used to obtain large numbers (many millions) of highly synchronized, optically transparent embryos that develop externally and are ideal for developmental studies, including studies that leverage live imaging or high-throughput analysis. Echinoderms also develop very rapidly and complete embryogenesis in as little as 2 days. Methods for embryo micromanipulation, fate mapping, lineage tracing, and analyzing and perturbing gene expression and function are available in multiple species within the phylum. A growing number of chromosome-level genome assemblies are available, which sample the phylogenetic diversity of the phylum. Echinoderm genomes are relatively compact when compared with the genomes of other deuterostomes, including vertebrates, as they have not undergone whole-genome duplications.

Studies with sea urchins and other echinoderms have contributed significantly to our understanding of developmental mechanisms. Echinoderms have a long history of contributions to fertilization biology (reviewed in Briggs and Wessel 2006) and

continue to be an important model system for the study of this biological process (Wessel and Wong 2009; Chassé *et al.* 2019; Meaders and Burgess 2020; Wozniak and Carlson 2020; Carlisle and Swanson 2021). Furthermore, echinoderm embryos are ideally suited to high-resolution, in vivo imaging, an approach that has been leveraged to analyze diverse developmental processes (Martik and McClay 2017; Henson *et al.* 2019, 2021; Spurrell *et al.* 2023). Availability of the full genomic sequences of several echinoderms has spurred insights regarding cell signaling processes in early development and their role in embryonic patterning (Wessel and Wong 2009; Byrne *et al.* 2015; Sun and Ettensohn 2017; Annunziata *et al.* 2019; Chiaramonte *et al.* 2020; Meaders and Burgess 2020; Carlisle and Swanson 2021; Chessel *et al.* 2023; Perillo *et al.* 2023), neural development and regeneration (Slota and McClay 2018; Slota *et al.* 2019; Byrne 2020; Alicea-Delgado and García-Arrarás 2021; Medina-Feliciano and García-Arrarás 2021; Piovani *et al.* 2021; Wolff and Hinman 2021; Czarkwiani *et al.* 2022; McClay 2022; Meyer and Hinman 2022; Zheng *et al.* 2022), germ cell specification (Fresques and Wessel 2018; Foster *et al.* 2020; Massri *et al.* 2021; Perillo *et al.* 2022), and responses to environmental stress (Martino *et al.* 2017; Ragusa *et al.* 2017; Garrett *et al.* 2020; Masullo *et al.* 2021; Chiarelli *et al.* 2022) and toxicity (Gökirmak *et al.* 2012; Shipp and Hamdoun 2012;

Nesbit *et al.* 2019; Li *et al.* 2020; Nogueira *et al.* 2021; Vyas *et al.* 2022; Perillo *et al.* 2023). The development and evolution of cell types is being transformed by the new technology of single-cell and single-nucleus RNA sequencing of multiple echinoderm species (Foster *et al.* 2020, 2022; Perillo *et al.* 2020; Massri *et al.* 2021; Paganos *et al.* 2021; Meyer *et al.* 2022, 2023; Satoh *et al.* 2022). Another new technology, RNA tomography, in combination with in situ hybridization, has been used to analyze patterns of gene expression in exquisite detail (Formery *et al.* 2023).

The first detailed animal gene regulatory network (GRN) model for development emerged from studies on sea urchins (Davidson *et al.* 2002a, 2002b). This work heralded the beginning of a new and growing research role for sea urchins and is still widely cited today (Verd *et al.* 2019; Rothenberg and Göttgens 2021; Day *et al.* 2022). Studies on GRNs were tremendously augmented by the first echinoderm genome sequence (Sea Urchin Genome Sequencing Consortium *et al.* 2006). Current work in this important field is expanding our understanding of the architecture of GRNs (Hinman *et al.* 2003, 2009; Peter and Davidson 2009, 2010, 2017; Damle and Davidson 2011; Andrikou *et al.* 2015; Dylus *et al.* 2016; Erkenbrack 2016; Erkenbrack *et al.* 2018; Fernandez-Valverde *et al.* 2018; Shashikant *et al.* 2018a, 2018b; Wang *et al.* 2019; Khor and Ettensohn 2022), GRN evolution (Erkenbrack *et al.* 2016; Israel *et al.* 2016; Khor and Ettensohn 2017; Khadka *et al.* 2018; Cary *et al.* 2020; Hogan *et al.* 2020; Hatleberg and Hinman 2021; Yamazaki *et al.* 2021; Ben-Tabou de-Leon 2022; Levin *et al.* 2022), linkages between GRNs and tissue morphogenesis (Rafiq *et al.* 2012; Annunziata *et al.* 2014; Martik and McClay 2015; Khor and Ettensohn 2022; Satoh *et al.* 2022; Tarsis *et al.* 2022), the regulation of GRNs by intercellular signaling pathways (Cui *et al.* 2014; Sun and Ettensohn 2014; Range 2018; Tsironis *et al.* 2021), and the utility of GRNs for reengineering cell specification (Damle and Davidson 2012; Pieplow *et al.* 2021). Based on these and other studies, sea urchins and other echinoderms remain a preeminent model for the analysis of developmental GRNs. At the same time, this work has pioneered the application of systems and GRN approaches to many other model organisms (Zmasek *et al.* 2007; Kubo *et al.* 2010; Dutkowski and Ideker 2011; Sánchez Alvarado 2012; Zmasek and Godzik 2013; Verd *et al.* 2019; Parker and Krumlauf 2020; Krumlauf and Wilkinson 2021; Rothenberg 2021; Rothenberg and Göttgens 2021; Day *et al.* 2022; Papadogiannis *et al.* 2022).

Adult echinoderms have potent, nonadaptive immune systems that utilize hundreds of receptor classes, including toll-like receptors (Buckley and Rast 2015). The study of echinoderm immunobiology has the potential to inform our understanding of the function and evolution of vertebrate immune systems (Smith and Davidson 1994; Buckley and Rast 2015, 2017, 2019; Reinardy *et al.* 2016; Buckley *et al.* 2017, 2019; Chiaramonte *et al.* 2019). The extraordinary capacity that echinoderms have for regeneration and the exceptional longevity of some species in the phylum are being leveraged to investigate the molecular and cellular mechanisms that underlie these characteristics (Byrne 2020; Alicea-Delgado and García-Arrarás 2021; Medina-Feliciano and García-Arrarás 2021; Piovani *et al.* 2021; Wolff and Hinman 2021; Czarkwiani *et al.* 2022; Meyer and Hinman 2022).

Echinobase is the central repository for a breadth of information that supports the international community of researchers who work with echinoderms. Users have easy access to the relational database and a user-friendly, intuitive interface for rapid access to functional genomics data, reagents, protocols, literature, developmental ontology, and community contacts. FAIR data management principles are implemented to make data findable, accessible, interoperable, and reusable (Wilkinson *et al.* 2016). Biocuration manually incorporates information that cannot be added automatically, including gene naming, genome annotation, reagent connection to genes, and data assignment to the Echinoderm Anatomical Ontology (ECAO) stages. Bioinformaticians work to make echinoderm data accessible to the broader biomedical community through genomic and gene expression analyses and comparisons.

## Overview of Echinobase content and usage

Echinobase (www.echinobase.org) has its origins in previous genomic databases, SpBase [2007–2012 (Cameron *et al.* 2009)] and EchinoBase [2012–2018 (Cary *et al.* 2018)]. The current version of the knowledgebase was developed as a clone of Xenbase, the *Xenopus* Model Organism Knowledgebase (Arshinoff *et al.* 2022; Fisher *et al.* 2023).

Echinobase provides search functionality for genes, the research community (people, labs, and organizations), publications, diseases, developmental anatomy, and Gene Ontology (GO) terms via the landing page search bar or pull-down menus. It currently contains 38,000 gene pages. Gene pages display gene, mRNA and protein models, Human Genome Organization Gene Nomenclature Committee–compliant names, diseases associated with the gene, multispecies orthology (including human, mouse, rat, zebrafish, chicken, and fruit fly), GO terms, a link to the JBrowse genome browser, publications on the gene, and any available reagents, including morpholinos, antibodies, and guide-RNAs (gRNAs). Tabs on the Summary gene page provide deeper gene-specific literature, transcript sequences, expression data, protein sequences, and protein–protein interaction predictions (based on human protein data).

Automated literature collection has retrieved over 18,000 publications for automated and manual curation. A preliminary version of an ECAO has been developed with standardized anatomy terms for developmental stages and parts that have been organized into a hierarchy with a visualization tool to graph the relationships between anatomical structures as they develop. To support the community, collections of data, protocols, and other resources are shared using a wiki (EchinoWiki) and a download site (download. echinobase.org). To enable interdisciplinary and collaborative studies, research, descriptions, and contact information of community members and groups are available and searchable.

Echinobase currently hosts genomes from species belonging to 3 of the 5 classes of echinoderms; 3 echinoids (sea urchins), 3 asteroids (sea stars), and 1 crinoid (feather star), with the less widely studied holothuroids (sea cucumbers) and ophiuroids (brittle star) yet to have genomes of sufficient quality to be supported. A unique feature of Echinobase is the visualization of multiple species orthology for fully supported species, including 2 sea urchins (*Strongylocentrotus purpuratus* and *Lytechinus variegatus*) and 2 sea stars (*Patiria miniata* and *Acanthaster planci*). These species have their genomes integrated into Echinobase's gene pages. Partially supported species do not appear on gene pages but have their genomes available on JBrowse and have BLAST capabilities; these species include *L. pictus* (a euechinoid sea urchin), *Asterias rubens* (a sea star), and *Anneissia japonica* (a feather star).

The current production and testing software environments are running in a private cloud at the University of Calgary where Xenbase is hosted and implemented as a fully virtualized federation of virtual machines (Karimi and Vize 2014). The private cloud is powered by 2 Lenovo x3850 M6 servers with 48 CPU cores and 1 TB of RAM each. Disk space is all-flash, contributing to high-speed retrieval of information from the database. VMware

vSphere serves as the hypervisor and provides load balancing and fault tolerance. All the infrastructure is behind a secure firewall and accessible only inside a VPN. Public access to the knowledgebase is via a standard Apache HTTPS web server. User statistics gathered by Google Analytics show 25–50 users per weekday and an international group of users. The main use of the site is to search for genes and access gene pages.

Since Echinobase was last reviewed (Arshinoff *et al.* 2022), several new features and upgrades have been added that have substantially improved the knowledgebase:

- The Search Gene function has been upgraded.
- *L. variegatus* (the green sea urchin) is fully supported.
- LOC (gene locus) symbols have been updated for 1:1 orthologs.
- Morpholino and gRNA alignments have been added to gene pages and JBrowse.
- PhylomeDB links have been added to gene pages.
- InterPro gene search is supported.
- KEGG (Kyoto Encyclopedia of Genes and Genomes) pathway names have been added.
- *L. pictus* (the painted sea urchin) is partially supported.
- JBrowse tracks have been added for functional genomic data visualization.
- The former version of the database (Legacy EchinoBase) has been retired, and data are now available at download. echinobase.org/echinobase/Legacy/.
- Code has been updated to improve speed and reliability, including the migration of code from IBM's WebSphere to the open-source Apache Tomcat.

## Navigating Echinobase

A News carousel and Announcements on the Echinobase landing page (https://www.echinobase.org/) provide current notices for the community regarding news and Echinobase content (Fig. 1). The landing page provides rapid access to information and is designed to allow intuitive navigation with multiple redundant paths for users to interact with the content (Fig. 1). For example, to quickly search for a gene of interest, the Search bar can be used directly as it defaults to gene searches. If a more detailed search is desired, the Gene Search menu item can be used. The Genes and Expression tile block on the landing page also has a link to Search for genes (Fig. 2). The Tutorial Videos (https://www.echinobase.org/echinobase/static-echinobase/HowTo.jsp) provide users with an overview of Echinobase, short step-by-step instructions for using JBrowse, and videos highlighting some of the features of Echinobase.

There are drop-down menus and tile blocks to access genomic content, including Genes and Expression data, Genomes, and BLAST. The ECAO anatomy terms can be searched directly from the search bar or accessed from the Anatomy and Development drop-down menu or the information block at the bottom of the landing page. Also on the menu bar are dropdowns that link to Resources, Literature, and Community searches and the Downloads menu. The latter provides access to Gene Page Reports and other data files for download.

## Echinobase gene pages

A gene search leads to a Summary page where various data types are collected and displayed (Fig. 3). Summary pages make extensive use of tabs to reduce clutter. At the top of each Summary page, the symbol, name, synonyms, and function of the gene are shown, followed by protein information and links. A graph displays transcript levels across developmental time for *S. purpuratus* (Tu *et al.* 2014). The main table on the gene page provides detailed information regarding gene ID and location, sequence visualization and BLAST functionality, a JBrowse genome browser link, and additional information and links regarding orthology, publications, and curated reagents (antibodies, morpholinos, and gRNAs). Additional tabs provide data regarding Expression, Gene Literature, GO Terms, Nucleotides (genomic and mRNA), Proteins, Interactants (literature cocitation), and a Wiki tab for miscellaneous notes regarding the gene.

## Multispecies integration

Echinobase uses *S. purpuratus* as the anchor species for orthology, as this species is more widely used for experimental studies than any other echinoderm and has a high-quality genome assembly. Orthology between *S. purpuratus* and humans is used to assign gene symbols and names. Currently, there is a gene page for every *S. purpuratus* gene model, and if there is a 1:1 ortholog in another species, then the tabulated information for that species is included. Orthology is determined using a DIOPT (Drosophila RNAi Screening Center Integrative Ortholog Prediction Tool)-based comparison between humans and *S. purpuratus* (Foley *et al.* 2021). Protein sequences are analyzed using several tools, and a minimum threshold of 3 tools was selected for identifying orthologs. Seven tools were used to compare *S. purpuratus* and human proteins for determining gene names and gene symbols; this approach identified 3,617 one-to-one orthologs. Five tools were used to compare *S. purpuratus* proteins with those of other echinoderms, and this lateral mapping was used to assign orthologs in other echinoderms (Fig. 3). This approach identified several thousand orthologs in *L. variegatus* (6,577), *P. miniata* (5,936), and *A. planci* (6,247). In the future, the display will be expanded to include genes with one-to-many and many-to-one relationships with human genes. Paralog analysis for *S. purpuratus* (i.e. genes duplicated in urchins relative to a human reference gene) has been completed, and the display of these paralogs on the gene pages is under development.

## Genomes, full and partial support

Echinobase only hosts genomes that have been submitted and annotated by NCBI. Experimentally relevant species and high-quality genome assemblies are hosted with 2 levels of support, full and partial. Full support involves full genome integration in the database, including gene pages, as well as BLAST, browsing via JBrowse, and inclusion in weekly data download reports. Full support represents a significant resource commitment and therefore must be approved by the Scientific Advisory Board. Partial support has BLAST, JBrowse, and genomic data download support but no gene page integration making this integration a much more rapid, and a less resource-dependent, process. The new *S. purpuratus* v5.0 (purple sea urchin), *P. miniata* v3.0 (bat star), and *L. variegatus* v3.0 (green sea urchin) genomes are available along with the *A. planci* v1.0 (crown-of-thorns sea star) genome and are fully supported. The *A. japonica* v1.0 (feather star), *As. rubens* v1.3 (sugar star), and *L. pictus* v2.1 (painted urchin) genomes are partially supported (JBrowse, BLAST, and Download). Links to other echinoderm genomes available at NCBI are listed on the EchinoWiki as Links to Additional Echinoderm Datasets.

## Gene nomenclature

Naming of genes is a multistep process that follows guidelines prepared by the Echinobase Nomenclature Steering Committee

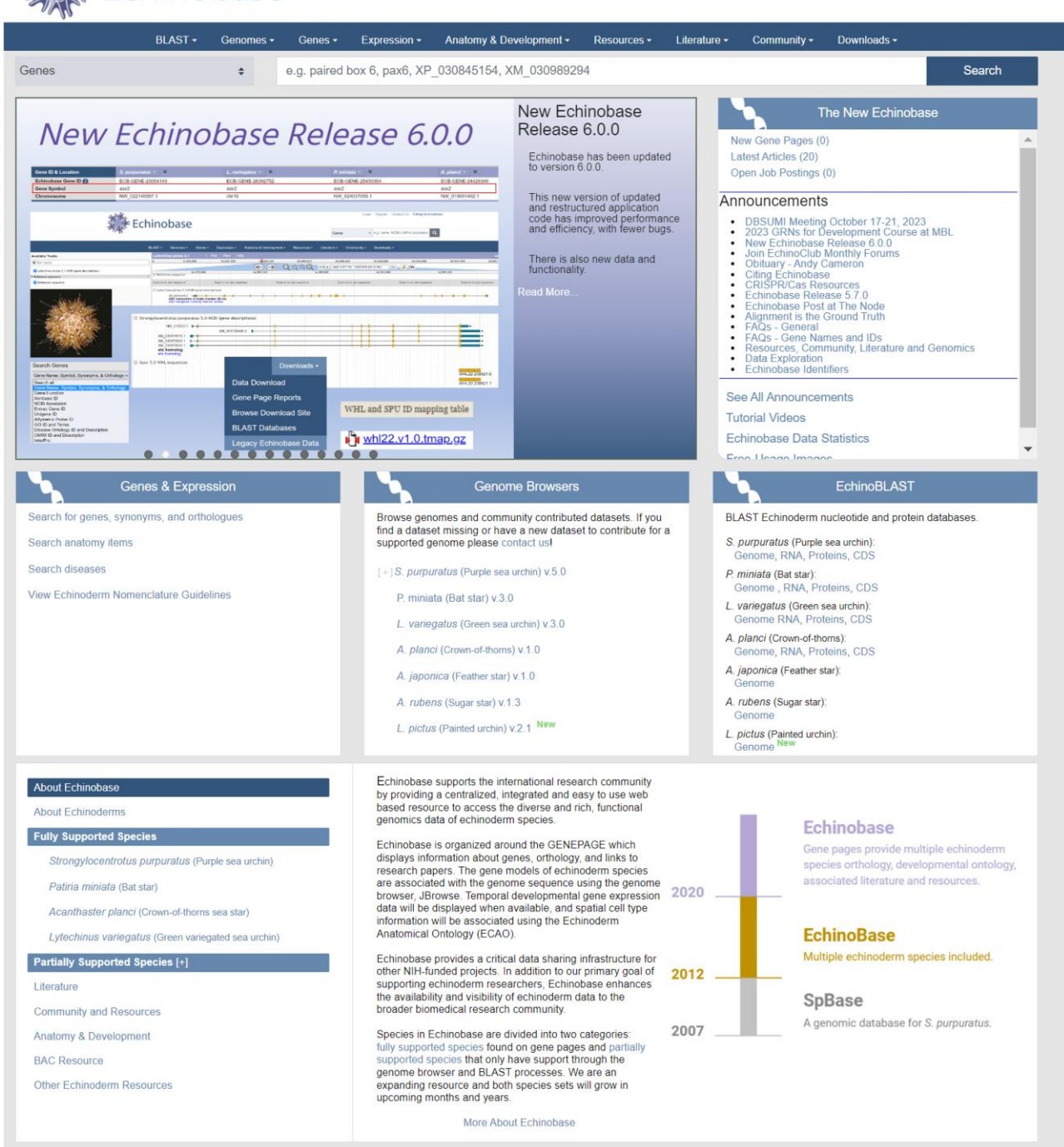

**Fig. 1.** The Echinobase landing page.

and Working Group (Beatman *et al.* 2021). Curators have manually assigned names to another set of genes that are prominent in the published literature, including those that are components of developmental GRNs. Echinoderm gene symbols and names are lowercase and italicized. For protein-coding genes, orthology is the determining factor; if there is no human ortholog, then a hierarchy has been established for assigning the appropriate name (https://www.echinobase.org/echinobase/static/gene/geneNomenclature.jsp). Currently, this is a manual process. When the gene symbol remains a LOC ID, the gene name that was assigned by

NCBI is available to provide some information regarding gene function.

## Gene expression, regulation, and manipulation

Our understanding of the evolution and function of genomes and genes is improved by providing high-quality, accurate genome assemblies, ortholog identification, and collection and annotation of transcriptomes and regulatory sequences, chromatin accessibility,

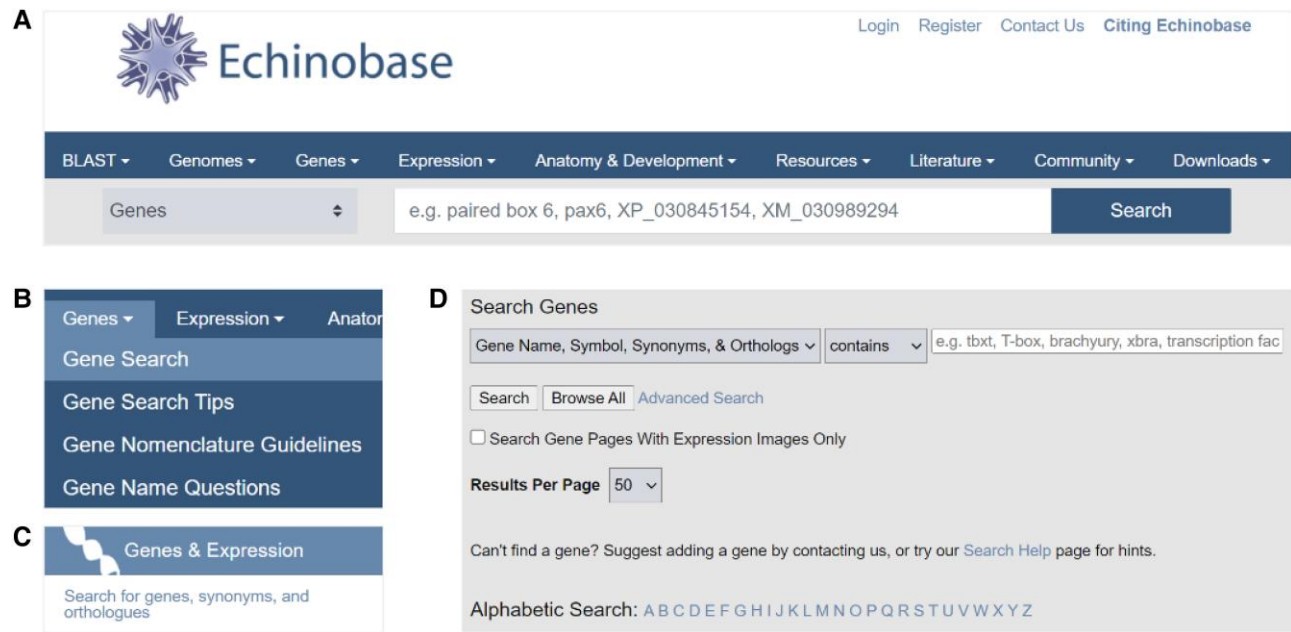

**Fig. 2.** Gene Search options from the landing page. A) Direct search using the search bar. B) Gene Search using the drop-down menu. C) Search for genes link from the Genes & Expression block. D) Both the drop-down and the link take users to the Search Genes page.

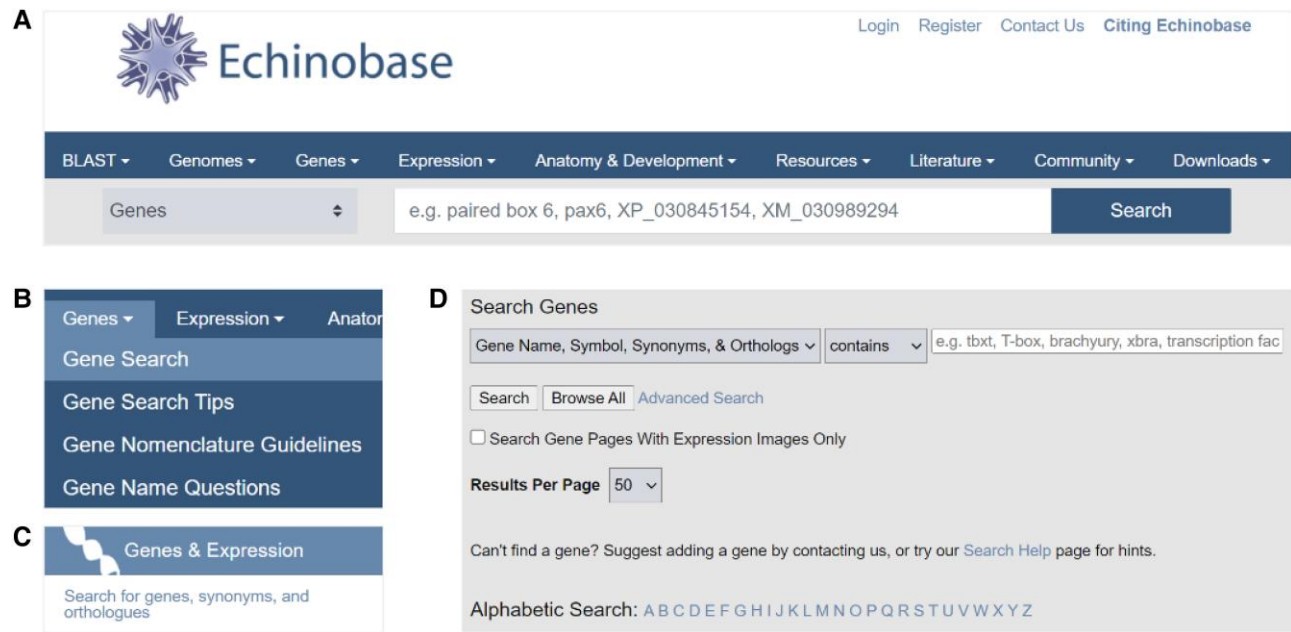

**Fig. 3.** A sample gene page shows all of the information gathered and displayed on the Summary tab. Species-agnostic information such as gene symbol, synonyms, molecular function, and disease associations are displayed at the top, while species-specific content is arranged in vertical columns below this more general content. Other tabs can be used to access more specific supporting information. Note that the gene in this example is orthologous *S. purpuratus*, *L. variegatus*, *P. miniata*, and *A. planci*. The species displayed, and the order in which they are displayed, can be set by users using the dropdowns on top of each column.

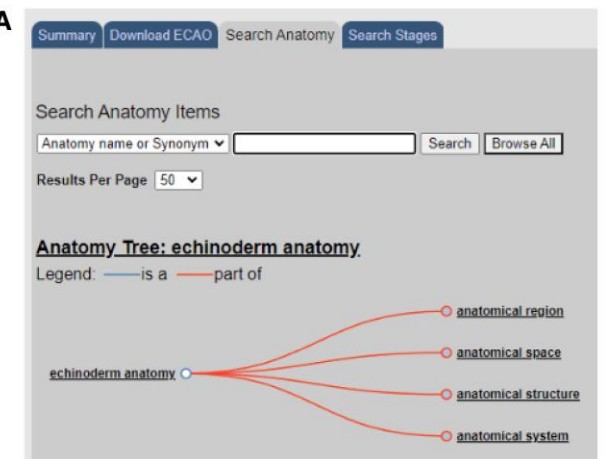
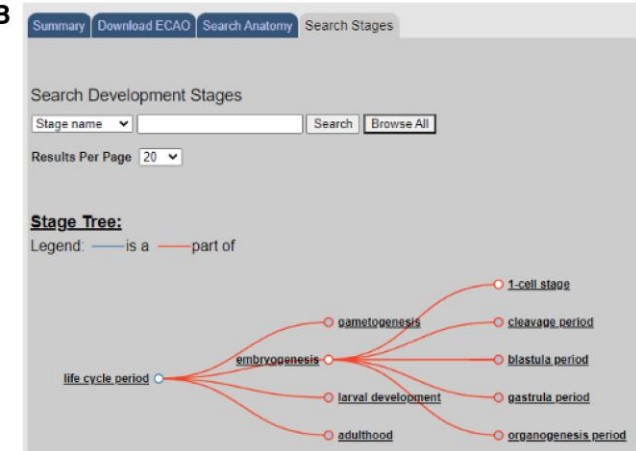

**Fig. 4.** The ECAO serves as a scaffold that can have data attached to it using spatial and temporal dimensions. A) The anatomy can be searched, and items are organized in a hierarchy of spatial structures. B) The developmental stages are arranged in a temporal hierarchy.

and other functional genomics datasets. Developmental RNA-seq data from *S. purpuratus* (Tu *et al.* 2014) are displayed as TPM (transcripts per million) vs time (in hours post-fertilization) plots on the gene pages. RNA-seq and ATAC-seq developmental expression datasets can be visualized in JBrowse as tracks or downloaded. As more data become available at NCBI, it will be updated on Echinobase. Recent additions for *S. purpuratus* include DNA methylation tracks ordered by stage, updated transcription factor–binding site tracks with linkouts to the CIS-BP database for all motifs, whole embryo ATAC-seq across different stages of *L. variegatus*, and RNA-seq tracks across different stages of *L. pictus*. Additionally, a JBrowse track was created to align historic *S. purpuratus* transcript models, which are identified by the prefix, WHL.

Users are also invited to submit data directly to Echinobase for incorporation as a JBrowse track. Bigwig file format is optimal, with identifiers that match the current JBrowse coordinates. Genome-wide maps of enhancer RNAs across multiple developmental stages are displayed as JBrowse tracks and are useful for enhancer identification and analysis (Khor *et al.* 2021). Identification and testing of *cis*-regulatory elements (Khor and Ettensohn 2022) and inducible gene expression methods (Khor and Ettensohn 2023; Zueva and Hinman 2023) allow for more detailed analysis of genomic elements involved in the regulation of transcription.

## Anatomy resources

The current ECAO describes anatomical entities and developmental stages of *S. purpuratus*, the most widely studied echinoderm. Curators use the ontology to label data in a consistent manner. The ECAO is searchable and relates each developmental stage to the stages that immediately precede and follow it (Fig. 4). Hierarchical structural and developmental relationships among anatomical entities are also included, as are the developmental stages when each entity is present.

## Resources, literature, community, and downloads

The Echinobase Resources serve to support the community by collecting data, protocols, and other resources that are then shared using the EchinoWiki and Download site. The EchinoWiki has a

wide range of information, including protocols and reagents. Antibodies and morpholinos are manually curated and displayed at the end of the gene page allowing for the rapid identification of gene-specific reagents for use in experiments. For genome-editing experiments, gRNAs that have been published are listed on gene pages and CRISPR (Clustered Regularly Interspaced Short Palindromic Repeats) methods are available on EchinoWiki.

The downloads available include all of the genomes and the Gene Page Reports. The literature associated with a set of echinoderm search terms has been collected (Karimi *et al.* 2021) and associated with both genes and tissues in the ECAO and will be updated weekly, making Echinobase a destination for relevant literature. Links in publication reports jump directly to gene pages, tissue descriptions, author pages, and more. In order to support the Community and to enable interdisciplinary and collaborative studies, research, descriptions, and contact information of community members and groups are available and searchable. Echinobase also supports the posting of available relevant job positions.

## Future directions

Echinobase is a critically important source of genomic information related to echinoderms. Without it, most of the important research resources that have been developed over the past decade (including the genome sequences themselves) would be almost useless. The continual improvement of this vital resource is therefore of the highest priority. Echinobase has unique and important roles in: (1) providing a paradigm for the integration of diverse types of biological data across multiple species and large evolutionary time scales in a single knowledgebase and (2) supporting GRN biology, including the use of multispecies data to study GRN evolution. Echinobase also leads the way in working with highly polymorphic genomes, which are typical for many species with large population sizes. Echinobase therefore serves as a technical resource to scientists establishing new organisms for genomic studies. As a medium-sized community, we are also well placed to test new approaches for data sharing. Echinobase therefore serves as an important resource for a wide community beyond those researchers who work on echinoderms.

Echinobase priorities are developed based on regular input from the research community, gathered in Town Hall Meetings and Surveys, and from the Scientific Advisory Board. Current priorities include the following.

## Genomes

A major goal is to add additional genome assemblies of species of echinoderms used as model organisms for genomic research. The genomes of echinoderms are large and polymorphic. Efforts to sequence and assemble them have often served as test cases for this kind of effort in general. The community has recently requested that annotated genomes of *L. pictus* and *Paracentrotus lividus* be made available on Echinobase. The *L. pictus* genome annotations are currently available with partial support.

## Gene annotations and ortholog identities

A central objective of many research programs in our community is to assay gene expression and function. Efforts should be made to generally improve gene annotations (e.g. splicing isoforms, noncoding RNAs, translation starts sites, and UTRs). This will globally improve the utility of Echinobase for all researchers and facilitate CRISPR gRNA design (Lin and Su 2016; Cui *et al.* 2017; Lin *et al.* 2019). A DIOPT-like pipeline has been developed for the ortholog prediction of *S. purpuratus* to humans (for gene names) and to other fully supported echinoderm species on Echinobase. This analysis will be expanded to include PhylomeDB phylogenetic analysis. GO terms will be expanded to improve gene expression studies, and visualization tools will be incorporated for the display of data.

The number of user-generated echinoderm transcriptomes is ballooning. There are currently 292 echinoderm transcriptomes in the NCBI sequence read archive representing 81 species. Most of these were collected for an explicit experimental purpose, and no consolidation has been undertaken. Thus, a huge amount of data is lost to the experimentalist. A further goal therefore is to collate transcriptomes from these many sources to provide high-quality reference transcriptomes from multiple species and, where possible, provide details of time points and tissue types.

## Echinoderm embryo ontology

The ECAO is currently being expanded from *S. purpuratus* to include 2 additional, widely used euechinoids (*L. variegatus* and *P. lividus*), a representative (and the most widely studied) cidaroid (*Eucidaris tribuloides*), and the bat star (*P. miniata*). The integration of the developmental stages and anatomical features of diverse echinoderm species into a single, unified, developmental ontology (the Echinoderm Embryo Ontology) will provide a unique and powerful tool for the curation of diverse, gene-related information across the phylum and the comparative analysis of data from multiple species.

## Resource sharing

Echinobase serves as a centralized resource for sharing protocols, reagents, and community news. Therefore, the knowledgebase must remain current. The EchinoWiki is available for posting protocols and validated reagents. Antibodies, morpholinos, and gRNAs are continuously curated and incorporated into gene pages as these reagents appear in publications. Links to gRNA tools will also be provided. Images are curated from articles and an image gallery is available. The Announcements on the landing page serve to notify the community of events and news.

## Genome-wide curation of regulatory sequences

The community is very interested in the genome-wide curation of noncoding, transcriptional regulatory sequences for many species, as determined from ATAC-seq profiles and other genomic signatures. Echinoderms are famous for the ease with which synchronous embryo cultures can be obtained, suiting them perfectly for the developmental profiling of chromatin architecture. Most importantly, echinoderm embryos are unusually well suited for functional *cis*-regulatory analyses of gene expression, an essential component of GRN studies. Such data are emerging from many labs, and it will be crucial to provide genome browser tracks or other portals to these data on Echinobase. This will greatly facilitate improved annotations of functional, noncoding DNA and the use of echinoderms for regulatory functional genomics. Such data are also routinely needed by researchers from other model systems and, in particular, the growing body of researchers performing comparative functional genomics that would like to use this major phylum in their analyses.

## Single-cell RNA-seq

A major goal is to further incorporate gene expression data, including single-cell RNA-seq data, into Echinobase. New endeavors to include spatial and quantitative expression should be included for *S. purpuratus* and other important experimental species. Significant individual lab efforts are directed at identifying spatial and quantitative gene expression profiles that can benefit the community as a whole. Providing these data in a format that can be readily accessed and cross-referenced from multiple species will aid comprehensive syntheses of GRN analyses, including for researchers from other communities. These should, as much as possible, also follow standards for other model systems outside of the echinoderms, to facilitate broader accessibility. Controlled vocabularies for developmental anatomy, and developmental stages, should be developed with the intent of coordinating with other taxa.

## Maintenance and updating

As the types and quantity of data expand, it becomes imperative to remodel the Echinobase web information system to ensure that it remains easily accessible to researchers, regardless of their experience with echinoderms. This will include the use of uniform nomenclature and searching tools, as well as intuitive links to external resources and databases. Efforts will be made to seek input from researchers in other systems and, in particular, other genomic web resource developers, to stimulate outreach efforts to service a broader community. The goal is to increase the impact of Echinobase and echinoderm research and to ensure that researchers from other communities can take advantage of the work done using echinoderms. These recommendations address critically important needs identified by the community, seek to make the best use of current resources, and are directed at enhancing the unique strengths of the echinoderm model system for the coming decade.

## Education and outreach

We have produced a number of papers and tutorial videos to help our users in accessing information on Echinobase. The availability of sea urchin gametes and the ease of their manipulation have made the sea urchin a popular source of educational material for many years. There are 2 widely used and complementary educational websites. "Sea Urchin Embryology" (https://depts.washington.edu/embryology/) provides essential information concerning animal procurement and handling, gamete collection, and fertilization, as well as detailed protocols for simple wet-lab exercises related to fertilization and early development. "Virtual Urchin" (https://depts.washington.edu/vurchin/) supports unique, interactive, web-based educational modules related to sea urchin development, including a virtual lab bench for simulating complex experimental manipulations. "Embryology Experiment"

kits are commercially available from Carolina Biological Supply Company and Gulf Specimen Marine Lab, attesting to the widespread use of sea urchin gametes and embryos as educational materials.

## Data availability

All data in this article are available at https://www.echinobase.org or at the given URLs.

## Funding

The development of Echinobase is funded by the Eunice Kennedy Shriver National Institute of Child Health and Human Development (NICHD), Grant No. P41 HD095831, awarded to VFH, CAE, and PDV.

## Conflicts of interest

The author(s) declare no conflicts of interest.

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

*Editor: T. Harris*