## [Peer Review File · Genetics]

Echinobase: a resource to support the echinoderm research community.

Cheryl Telmer, Kamran Karimi, Macie Chess, Sergei Agalakov, Bradley Arshinoff, Vaneet Lotay, Dong Zhuo Wang, Stanley Chu, Troy Pells, Peter Vize, Veronica Hinman, and Charles Etensohn

NOTE: The reviews and decision letters are unedited and appear as submitted by the reviewers.

In extremely rare instances and as determined by a Senior Editor or the EIC, portions of a review may be redacted. If a review is signed, the reviewer has agreed to no longer remain anonymous.

The review history appears in chronological order.

Review Timeline:

Submission Date:	2023-11-13
Editorial Decision:	2023-12-11
Revision Received:	2023-12-21
Accepted:	2023-12-27

December 11, 2023
RE: GENETICS-2023-306648

Dear Dr. Ettensohn:

I am pleased to accept your manuscript entitled "Echinobase: a resource to support the echinoderm research community." for publication in GENETICS, pending minor revision.

Please submit your revision along with a response to the reviewers' concerns and suggestions, which can be viewed at the bottom of this email. As described in Review #1, consider expanding the scope of the user community to be more inclusive, and describe in more detail the decision criteria for full and partial support species (a point of interest not just to end users of Echinobase but of operators of other model organism databases). I expect this can be done within 30 days.

Upon resubmission, please include:

1. A clean version of your manuscript;
2. A marked version of your manuscript in which you highlight significant revisions carried out in response to the major points raised by the editor/reviewers (track changes is acceptable if preferred);
3. A detailed response to the editor's/reviewers' comments and to the concerns listed above. Please reference line numbers in this response to aid the editors.

Additionally, please ensure that your revision is formatted for GENETICS: <https://academic.oup.com/genetics/pages/general-instructions>.

Follow this link to submit the revised manuscript: Link Not Available

Thank you for submitting your research to Genetics.

Sincerely,

Todd Harris
Associate Editor
GENETICS

Approved by:
Paul Sternberg
Senior Editor
GENETICS

Reviewer comments:

Reviewer #1 (Comments for the Authors (Required)):

This manuscript documents the continued presence of a knowledgebase for the echinoderm community using Xenbase as the template. It is nicely written and documented. It has, however, been described several times - as documented in Echinobase citations.

What is new since the last manuscript describing Echinobase? Arshinoff et al., NAR 2022 and the several published in 2021? And it is a website that should be self-sufficient.

It would be very helpful to provide updates of Echinobase - not just a re-description of the site. It is being presented wholly, which is valuable, but for readers having used Echinobase versions before, it is unfortunate to make them wade through the same verbiage looking for improvements/additions. A listing of main upgrades would help greatly.

Abstract - why "in particular" and limit emphasis of this phylum to developmental studies and gene regulatory networks? These animals have much more to contribute and the Abstract would be a good place for this fact. Were I not interested in GRNs and saw the description of this site, I would move on.

Why is there a two-tiered hierarchy? Supported and partially supported? Is this an Echinobase thing or quality of genomic resources thing? The reasoning is not explained herein.

I do not see the word "chromosome" in this manuscript. Curious in such a manuscript submitted to such a journal. Where is the display of synteny between these species? It seems like these features would be helpful given the GRN emphasis.

Reviewer #2 (Comments for the Authors (Required)):

This manuscript describes Echinobase, a genomic resource for the community of investigators who focus their research on Echinoderms. The resource is similar to those for other embryo models, especially *Xenopus*, since one of the authors is strongly involved in establishing and furthering the resource for that model embryo. The content is dry reading and reminds me of Joe Friday on that old show "Dragnet": "Just the facts ma'am". The facts describe the content that can be found in Echinobase. Rather than make suggestions on change, I suspect that 10 reviewers would make 10 different suggestions and still the content would remain much the same. Instead, I will make one comment on the latter component of the manuscript. I think the aspirations included in the text are excellent. They will be incomplete since changes in technology will dictate changes in aspiration and inclusion, but inclusion of those ideas are important because it indicates the Echinobase team and their advisors want to make this resource a living website that changes and advances with the field rather than simply be archival.

Associate Editor comments:

This manuscript is a solid description of the contents and implementation of Echinobase. Despite reviewers comments of the material being dry, this provides a useful introduction to the website for new users. I am curious about the choice of using an on-premise cloud for hosting when commercial cloud services offer extensive tools for devops, security, and storage and (at least) cost parity. I'd suggest that a future directive should also include due diligence exploration of cloud offerings for greater interoperability with other MODs and more robust long-term maintenance.

Subject: GENETICS-2023-306648 - Editorial Decision

December 11, 2023

RE: GENETICS-2023-306648

Dear Dr. Ettensohn:

I am pleased to accept your manuscript entitled "Echinobase: a resource to support the echinoderm research community." for publication in GENETICS, pending minor revision.

Please submit your revision along with a response to the reviewers' concerns and suggestions, which can be viewed at the bottom of this email. As described in Review #1, consider expanding the scope of the user community to be more inclusive, (see below, added text Line 19-20) and describe in more detail the decision criteria for full and partial support species (a point of interest not just to end users of Echinobase but of operators of other model organism databases) (see below, addressed in Lines 278-290). I expect this can be done within 30 days (yes).

Upon resubmission, please include:

1. A clean version of your manuscript;
2. A marked version of your manuscript in which you highlight significant revisions carried out in response to the major points raised by the editor/reviewers (track changes is acceptable if preferred);
3. A detailed response to the editor's/reviewers' comments and to the concerns listed above. Please reference line numbers in this response to aid the editors.

Additionally, please ensure that your revision is formatted for GENETICS:.

<https://academic.oup.com/genetics/pages/general-instructions>.

Follow this link to submit the revised manuscript: Link Not Available

Thank you for submitting your research to Genetics.

Sincerely,
Todd Harris
Associate Editor
GENETICS

Approved by:
Paul Sternberg
Senior Editor
GENETICS

Reviewer comments:

Reviewer #1 (Comments for the Authors (Required)):

This manuscript documents the continued presence of a knowledgebase for the echinoderm community using Xenbase as the template. It is nicely written and documented. It has, however, been described several times - as documented in Echinobase citations.

What is new since the last manuscript describing Echinobase? Arshinoff et al., NAR 2022 and the several published in 2021? And it is a website that should be self-sufficient.

It would be very helpful to provide updates of Echinobase - not just a re-description of the site. It is being presented wholly, which is valuable, but for readers having used Echinobase versions before, it is unfortunate to make them wade through the same verbiage looking for improvements/additions. A listing of main upgrades would help greatly.

To address this concern we have added a list of new features and upgrades starting at Line 202-216.

Since Echinobase was last reviewed (Arshinoff *et al.* 2022), several new features and upgrades have been added that have substantially improved the knowledgebase:

- Search Gene upgraded
- *Lytechinus variegatus* fully supported
- LOC symbols updated for 1:1 orthologs
- Morpholino and gRNA alignments on gene pages and JBrowse
- PhylomeDB links on gene pages
- Support for InterPro gene search
- KEGG Pathway names added
- *Lytechinus pictus* (painted urchin) partially supported
- JBrowse tracks added for functional genomic data visualization
- Legacy EchinoBase has been retired and data are now available at download.echinobase.org/echinobase/Legacy/
- Code updated to improve speed and reliability including migration of code from IBM's WebSphere to the open source Apache Tomcat

We also added some details regarding the data available on JBrowse, Lines 310-314.

Recent additions for *S. purpuratus* include DNA methylation tracks ordered by stage and updated transcription factor binding site tracks with linkouts to the CIS-BP database for all motifs, whole embryo ATAC-seq across different stages of *L. variegatus* and RNA-seq tracks across different stages for *L. pictus*. Additionally, a JBrowse track was created to align historic WHL transcripts.

Abstract - why "in particular" and limit emphasis of this phylum to developmental studies and gene regulatory networks? These animals have much more to contribute and the Abstract would be a good place for this fact. Were I not interested in GRNs and saw the description of this site, I would move on.

We have added some text to the Abstract emphasizing additional contributions of echinoderm research.

Line 19-20 added "environmental, evolutionary and" and then removed "in particular" and replaced it with "including research on developmental"

Why is there a two-tiered hierarchy? Supported and partially supported? Is this an Echinobase thing or quality of genomic resources thing? The reasoning is not explained herein.

Lines 278-290 we added text to clarify the process for decision-making regarding species inclusion and support level on Echinobase. It now reads as follows (changes indicated in red).

Genomes, full and partial support

Echinobase **only** hosts genomes that have been submitted and annotated by NCBI. **Experimentally relevant species and** high quality genome assemblies are hosted with two levels of support, full and partial. Full support involves full genome integration in the database, including gene pages, as well as BLAST, browsing via JBrowse, and inclusion in weekly data download reports. **Full support** represents a significant resource commitment **and therefore must be approved by the Scientific Advisory Board**. Partial support has BLAST, JBrowse, and genomic data download support, but no gene page integration making this integration a much more rapid, **and a less resource dependent** process. The new *S. purpuratus* v5.0 (**purple sea urchin**), *P. miniata* (**bat star**) v3.0 and *L. variegatus* v3.0 (**green sea urchin**) genomes are available along with the *A. planci* v1.0 (**crown-of-thorns sea star**) genome and are fully supported. The *Anneissia japonica* v1.0 (feather star), *Asterias rubens* v1.3 (sugar star) and *Lytechinus pictus* v2.1 (painted urchin) genomes are partially supported (JBrowse, BLAST and Download). Links to other echinoderm genomes available at NCBI are listed on the EchinoWiki as Links to Additional Echinoderm Datasets.

I do not see the word "chromosome" in this manuscript. Curious in such a manuscript submitted to such a journal. Where is the display of synteny between these species? It seems like these features would be helpful given the GRN emphasis.

On Echinobase, chromosomes are indicated in JBrowse if they are provided with the assembly. We are currently discussing formalized chromosome nomenclature standards consistent with UCSC formatting, and plan to implement this in the GFF files we generate and export. When we display an annotation by a third party, we do not edit the author's annotation. We do plan to include synteny viewers in the near future, and are presently comparing different options including JBrowse2, UCSC and the MGI multigenome viewer. Links to test versions will be sent to the SAB for input once available.

Reviewer #2 (Comments for the Authors (Required)):

This manuscript describes Echinobase, a genomic resource for the community of investigators who focus their research on Echinoderms. The resource is similar to those for other embryo models, especially *Xenopus*, since one of the authors is strongly involved in establishing and furthering the resource for that model embryo. The content is dry reading and reminds me of Joe Friday on that old show "Dragnet": "Just the facts ma'am". The facts describe the content that can be found in Echinobase. Rather than make suggestions on change, I suspect that 10 reviewers would make 10 different suggestions and still the content would remain much the

same. Instead, I will make one comment on the latter component of the manuscript. I think the aspirations included in the text are excellent. They will be incomplete since changes in technology will dictate changes in aspiration and inclusion, but inclusion of those ideas are important because it indicates the Echinobase team and their advisors want to make this resource a living website that changes and advances with the field rather than simply be archival.

We agree with the comment that we want Echinobase to remain current and not only a repository for data.

Associate Editor comments:

This manuscript is a solid description of the contents and implementation of Echinobase. Despite reviewers comments of the material being dry, this provides a useful introduction to the website for new users. I am curious about the choice of using an on-premise cloud for hosting when commercial cloud services offer extensive tools for devops, security, and storage and (at least) cost parity. I'd suggest that a future directive should also include due diligence exploration of cloud offerings for greater interoperability with other MODs and more robust long-term maintenance.

Agreed, and we are in the process of designing a commercial cloud option, but as our servers are very high performance and our code relies on these considerable resources (e.g. 2 TB of RAM) moving is actually a major undertaking as a lot of code has to be refactored, especially SQL and MQTs. Replicating anywhere near our present resources would be extremely cost prohibitive, so a lot of redesign is involved. It is also complicated by application licensing- which will be resolved once we have moved to an all open-source environment, which is also in process. The present manuscript was written to highlight resources for potential users. We will write a more technical report on the migration process once it is complete, likely in around a year's time.

December 27, 2023

RE: GENETICS-2023-306648R1

Dr. Charles A. Ettensohn
Carnegie Mellon University

Dear Dr. Ettensohn:

Congratulations! We are delighted to inform you that your manuscript entitled "Echinobase: a resource to support the echinoderm research community." is acceptable for publication in GENETICS. Many thanks for submitting your research to the journal.

To Proceed to Production:

Add oupsupport@scipris.com and genetics.oup@novatechset.com (or the domains @scipris.com and @novatechset.com) to your email program's "safe senders" list. You will be contacted by both at various points during the production process.

1. Format your article according to GENETICS style, as discussed at <https://academic.oup.com/genetics/pages/general-instructions>. Ensure that you comply with data and community resource citation guidelines (<https://academic.oup.com/genetics/pages/general-instructions#Data-Policy>).
2. Upload your final files at <https://genetics.msubmit.net>.
3. Your currently-accepted manuscript (unedited, as submitted, reviewed, and accepted) will be published at GENETICS and deposited into PubMed as an Advance Access article. Notify sourcefiles@thegsajournals.org before signing your license if you do not wish to publish your article via Advance Access.
4. We invite you to submit an original color figure related to your paper for consideration as cover art. Please email your submission to the editorial office or upload it with your final files. You can submit a small-sized image for evaluation, and if selected, the final image must be a TIFF file 2513px wide by 3263px high (8.375 by 10.875 inches; resolution of 600ppi). Please avoid graphs and small type.
5. After files are sent to Oxford University Press we use SciPris to manage article licensing and payment. If you do not have a SciPris account, you will receive an email from no-reply@scipris.com to sign up to use Oxford University Press' author portal. After logging in, follow the online instructions to sign your licence and arrange any payment due.

If you have any questions or encounter any problems while uploading your accepted manuscript files, please email the editorial office at sourcefiles@thegsajournals.org.

Sincerely,

Todd Harris
Associate Editor
GENETICS

Approved by:
Paul Sternberg
Senior Editor
GENETICS